# Health service utilization and associated factors among fee waiver beneficiaries in Ethiopia: Systematic review and meta-analysis

Bizunesh Fantahun Kase[1]*, Hiwot Altaye Asebe[1], Etsay Woldu Anbesu[2], Dejen Kahsay Asgedom[1], Abubeker Alebachew Seid[3], Abdulkerim Hassen Moloro[4], Aragaw Asfaw Hasen[5], Nuru Mohammed Hussen[6], Molla Getie Mehari[7], Kassaye Getaneh Arge[6], Abdu Hailu Shibeshi[6]

1 Department of Public Health, College of Medicine and Health Sciences, Samara University, Samara, Ethiopia, 2 Department of Public Health, College of Medicine and Health Sciences, Samara University, Semera, Ethiopia, 3 Department of Nursing, College of Medicine and Health Sciences, Samara University, Semera, Ethiopia, 4 Department of Nursing, College of Medicine and Health Sciences, Samara University, Samara, Ethiopia, 5 Department of Statistics, College of Natural and Computational Sciences, Samara University, Semera, Ethiopia, 6 Department of Statistics, College of Natural and Computational Sciences, Samara University, Samara, Ethiopia, 7 Department of Medical Laboratory Science, Injibara University, College of Medicine and Health Sciences, Injibara, Ethiopia

* bezuneshf656@gmail.com

## Abstract

### Background

Health service utilization serves as a vital indicator of healthcare access and equity. In Ethiopia, the fee waiver system is a key component of healthcare financing reforms designed to improve access to essential health services for economically disadvantaged populations. However, the evidence regarding health service utilization among fee waiver beneficiaries remains inconsistent. This systematic review and meta-analysis synthesize existing studies to provide comprehensive insight on health service utilization and associated factors among fee waiver beneficiaries in Ethiopia.

### Methods

A systematic search of peer-reviewed articles and gray literature was conducted up to February 2024, in databases such as PubMed/MEDLINE, African Journals Online (AJOL), Cumulative Index to Nursing & Allied Health Literature (CINAHL), Science Direct, Research4life, and Google Scholar. A systematic review and meta-analysis were conducted in accordance with the PRISMA guidelines. Data were extracted using Microsoft Excel and analyzed with STATA 17 software. The quality of studies was assessed using Joanna Briggs Institute (JBI) checklists. The pooled prevalence of health service utilization among fee waiver beneficiaries was estimated using random-effects meta-analysis. Subgroup analyses were performed based on study

**Data availability statement:** ll relevant data are within the manuscript and its Supporting Information files.

**Funding:** The author(s) received no specific funding for this work.

**Competing interests:** The authors have declared that no competing interests exist.

**Abbreviations:** AJOL: African Journals Online; CINAHL: Cumulative Index to Nursing and Allied Health Literature; JBI: Joanna Briggs Institute; LMIC: Low- and Middle-Income Countries; MeSH: Medical Subject Headings; PRISMA-P: Preferred Reporting Items for Systematic Review and Meta-Analysis Protocols.

regions. Publication bias was evaluated with a DOI plot, the Luis Furuya Kanamori (LFK) index, and Egger's test, while heterogeneity was assessed using the I² statistic.

## Results

The study analyzed seven primary studies comprising a total of 11,488 participants. All the included studies demonstrated a low risk of bias, and no significant evidence of publication bias was detected among them. The pooled prevalence of health service utilization was found to be 60.57% (95% CI: 58.11–63.04; $I^2 = 54.2\%$, $p = 0.041$). A family size of fewer than five was negatively and significantly associated with health service utilization (OR = 0.69, 95% CI: 0.51–0.95; $I^2 = 0.0\%$, $p = 0.47$). On the other hand, having chronic diseases was positively and significantly associated with health service utilization among fee waiver beneficiaries (OR = 4.85, 95% CI: 1.34–17.56; $I^2 = 93.5\%$, $p < 0.001$). Residence showed no significant association (OR = 1.58; 95% CI: 0.03–71.49), with wide confidence intervals reflecting considerable uncertainty.

## Conclusion

The findings suggest that a significant number of beneficiaries accessed health services, indicating that the system is likely contributing to enhanced healthcare access for the target population. However, this also highlights the need for further efforts to ensure broader and more equitable utilization. The analysis reveals that health service utilization is negatively associated with a family size of fewer than five and positively associated with having chronic diseases. To improve the utilization rate among poor populations, policymakers in Ethiopia should implement integrated strategies that address these key factors and target barriers to healthcare access.

## Introduction

Access to health services is among the key factors that determine the overall health outcomes of individuals, by enabling them to receive necessary medical and preventive interventions timely [1]. So far, in rural and remote areas, challenges like geographical distance, inadequate transportation, and particularly financial barriers often hinder access [2]. However, access disparities also exist in urban contexts due to constraints such as high health-care expenditures, overcrowded facilities, and socioeconomic disadvantages among vulnerable populations, particularly slum dwellers [3]. Financial constraints are a major barrier to accessing healthcare, particularly for low-income individuals already struggling to meet basic needs [4,5]. Unaffordable costs of out-of-pocket expenditure for getting medical services are among the key barriers that individuals face when seeking healthcare, which leads to a high prevalence of preventable diseases, growing healthcare access disparities, and poorer overall health outcomes among the population [6]. In order to address these challenges and ensure equitable access across populations, it is essential to build need-based policy measures and a strong social support system [7]. In response to this,

a fee waiver program is developed as a key strategy to reduce these economic barriers, which then promotes progress toward universal health coverage (UHC) [8,9].

Evidence from several nations suggests that fee waiver schemes can improve health-care utilization. In this regard, a study conducted in South Korea among Medical Aid beneficiaries showed that the system led these beneficiaries to use inpatient and outpatient services more frequently than those with health insurance [10]. Similarly, a qualitative study conducted in Iran reported that the removal of user fees improves financial protection and increases health service utilization among the poor [11]. Other review articles from low- and middle-income countries have also indicated that waiving fees for health services improved utilization and suggested that fee removal should be part of broader health reform initiatives [12,13]. A scoping review in Africa showed that the elimination of user fees has a positive effect on health service utilization [14]. In Malawi, a quasi-experimental study revealed that eliminating user fees was associated with higher health-care utilization and an increase in infectious disease diagnoses [15]. Likewise, studies from Afghanistan and Uganda found that removing user fees increased the utilization of health services [16,17]. In India, the implementation of a fee waiver scheme at the tertiary level also contributed to overcoming financial barriers and improving access to healthcare for the poor [18].

On the other hand, a study from Burkina Faso found that a targeted user fee intervention was not effective in improving health service utilization among the ultra-poor. The findings suggested that, in addition to such strategies, it is also important to identify other significant barriers to health service uptake in the pursuit of achieving Universal Health Coverage (UHC) [19]. Similarly, a study conducted in South Korea also found no significant difference in medical service utilization between medical aid beneficiaries and individuals above the poverty line. The authors recommended that, policy interventions should consider factors influencing utilization, such as the presence of chronic diseases and perceived health needs [20].

A number of socio-economic and organizational factors were found to be significantly associated with health service utilization among fee waiver beneficiaries. These include perceived health status [10,21,22], household size [10,23,24], place of residence [10,21], presence of chronic diseases [10,21,25], level of education [23–25], income level [21,23], shortage of drugs and procedure [26,27], non-medical costs [22,25,26], perceived distance to health facilities [22].

Ethiopia initiated healthcare financing reform strategies in 1998, one of which was systematizing fee waiver system [28]. This initiative aims to minimize healthcare disparities and enhance overall health outcomes by focusing on the most disadvantaged and impoverished populations [29]. In many regions, a significant proportion of the population is eligible for and enrolled in fee waivers, which highlights the magnitude of the challenge [30]. Over the last few years, there has been a notable increase in government funding for fee waivers to improve healthcare access. While this is a positive move, the financing covers less than 10% of the country's poorest citizens [31]. In addition, despite national efforts, there are still major issues at the decentralized level. In many areas, fee waiver coverage is still limited, and their implementation varies. Local governments in several woredas are reluctant to implement fee waivers in their entirety because of the burden it would have on their already limited budgets [29].

Even though these initiatives have a lot of prospects, currently available evidence in Ethiopia is region-specific and inconsistent in its findings [21–27]. The absence of comprehensive evidence on this matter hinders the ability to identify the national estimate and generalize it to the larger community. As a result of the lack of pooled findings, policymakers and healthcare providers also face major challenges in the efforts they make to improve the effectiveness of this system. Therefore, the purpose of this study is to examine health service utilization among fee waiver beneficiaries and identify factors that influence their use in Ethiopia, thereby giving a thorough understanding to guide future policies and actions.

## Materials and methods

### Study setting

This study provides a comprehensive analysis of existing research on health service utilization among fee waiver beneficiaries in Ethiopia, along with the factors associated with their utilization.

## Protocol and registration

The protocol for this systematic review and meta-analysis is registered in the International Prospective Register of Systematic Reviews (PROSPERO) database with the ID number CRD42024575840.

## Information sources and search strategy

A comprehensive search of both academic and gray literature was performed up to February 2024. The databases utilized for our search included African Journals Online (AJOL), Google Scholar, Cumulative Index to Nursing & Allied Health Literature (CINAHL), Science Direct, Research4life, and PubMed/MEDLINE. The search string incorporated Medical Subject Headings (Mesh), keywords, and free text search terms such as "Health services," "utilization," "uptake," "fee waiver," "beneficiaries," and "recipient." These terms were combined using Boolean operators to create focused search queries, enabling us to identify studies relevant to our review. Detailed supporting information is available for the complete Medline/PubMed search strategy, as well as for other databases. (Supplemental file 1.)

## Eligibility criteria

This study included cross-sectional studies conducted in Ethiopia that examined health service utilization among fee waiver beneficiaries, using clear outcome measures. Both published and unpublished studies from peer-reviewed journals, regardless of language, were considered. There were no restrictions on publication dates, with all studies conducted up to 2024 being included. However, other types of works, such as letters, reviews, commentaries, conference proceedings, qualitative studies, editorial letters, case reports, and case series, were excluded. Each included study was confirmed to meet all predefined eligibility criteria before proceeding with data extraction and synthesis.

## Selection of studies

The Endnote X7 software reference manager was utilized to collect and organize search results, remove duplicate articles, and manage citations. The screening process involved two stages following an initial assessment of article titles. In the first stage, the first two authors (BFK and AHM) independently reviewed the titles and abstracts based on the eligibility criteria. In the second stage, abstracts that met the criteria underwent full-text screening. Only studies approved by both authors were included in the full review. Any disagreements were resolved through discussion and consultation with a third reviewer (AAS). Reasons for exclusion were documented for all excluded studies, resulting in a finalized list of studies for data extraction.

## Data extraction and management

Two independent reviewers (BFK & AHS) utilized a Microsoft Excel spreadsheet to extract relevant data after all qualifying studies were identified. The data extraction format was designed to be consistent with the Joanna Briggs Institute (JBI) guidelines for research syntheses and systematic reviews [32].

For each included article, the data extraction tool captured information such as the first author's last name, publication year, region, study setting, study design, sample size, response rate, the prevalence of health service utilization among fee waiver beneficiaries, factors associated with it, the effect size of risk factors (expressed as odds ratios), risk of bias results, name of the data extractor, and the date of data extraction. All relevant data were fully reported in the included studies; therefore, no data were missing for the outcomes of interest.

## Quality assessment

Two reviewers, BFK and HAA, independently evaluated the quality of the studies using the Joanna Briggs Institute (JBI) criteria for analytical cross-sectional studies [33]. The assessment was conducted with a checklist consisting of eight

parameters. (1) clear inclusion criteria in the sample; (2) detailed description of the study subject and study setting; (3) use of a valid and reliable method to measure exposure; (4) use of objective, standard criteria for measuring the condition; (5) identification of confounding factors; (6) development of strategies to deal with confounding factors; (7) use of a valid and reliable method to measure outcomes; and (8) use of appropriate statistical analysis [34].

The tool provides response options of yes, no, not applicable, and unknown. Responses marked as "yes" were assigned a score of one, while unclear, irrelevant, or absent responses received a score of zero. The total risk of bias score was categorized into three levels: low (6–8), moderate (3–5), and high (0–2). For this review, all the included articles were assessed to have a low risk of bias [35,36]. Any discrepancies between the two reviewers were resolved by the author (EWA) during the critical appraisal process.

### Outcome of interest

The primary objective of this review was to determine the overall prevalence of health service utilization among fee waiver beneficiaries, expressed as a percentage. The secondary objective was to determine the association between health service utilization among fee waiver beneficiaries and its predictors in Ethiopia, using the pooled odds ratio (OR) or effect size with a 95% confidence interval.

### Statistical methods and data analysis

The retrieved data was organized in a Microsoft Excel spreadsheet and subsequently exported to STATA version 17 for analysis. To quantify heterogeneity among the included studies, the index of heterogeneity (I² statistic) was applied. Thresholds of 25%–50%, 50%–75%, and >75% represented low, moderate, and high heterogeneity, respectively [37]. The pooled prevalence of health service utilization among fee waiver beneficiaries was estimated using a random-effects model, accounting for between-study heterogeneity [38].

A subgroup analysis was conducted to examine variations in the pooled prevalence of health service utilization across the study regions. Sensitivity analysis was performed to assess the influence of individual studies on the overall pooled estimate. Publication bias was evaluated using a DOI plot, the Luis Furuya Kanamori (LFK) index, and Egger's test. For Egger's test, a p-value of less than 0.05 was considered indicative of publication bias. Similarly, an LFK index value outside the range of −1–1 suggested the presence of publication bias. Factors associated with health service utilization among fee waiver beneficiaries were identified using a pooled odds ratio (OR) with a 95% confidence interval, with significance determined at a p-value of less than 0.05. The findings were presented using a combination of text, tables, and graphs.

### Ethical consideration

As this systematic review and meta-analysis did not involve the collection of primary data, ethical approval was not required. However, all included studies were carefully reviewed to confirm that they had obtained appropriate ethical clearances and reported informed consent from their participants.

## Results

### Search results

Initially, 515 studies were identified using various electronic databases such as PubMed, Google Scholar, CINHAL, African Journals Online, science direct, and research4life. Following the exclusion of 46 studies due to duplication, 456 studies were dropped based on titles and abstracts unrelated to the research topic. Subsequently, six studies were excluded based on the eligibility criteria. A list of all excluded studies, along with the specific reasons for their exclusion, is provided in (Supplementary file 2). Finally, this review included a total of seven studies. The findings were presented adhering to the Preferred Reporting Items for Systematic Review and Meta-analysis (PRISMA-2020) guidelines [39] (Fig 1).

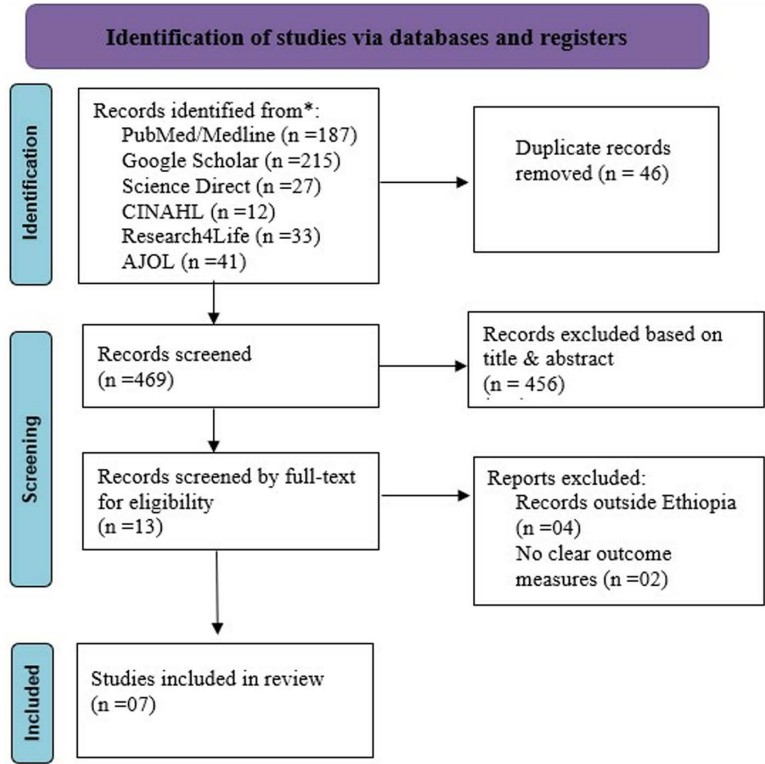

**Fig 1. RISMA flow diagram for the systematic review and meta-analysis of health service utilization among fee waiver beneficiaries and associated factors in Ethiopia, 2024.**

## Characteristics of the included studies

This review and meta-analysis included seven studies with sample sizes ranging from 163 to 681 participants, for a total of 3,440 individuals. Of these studies, three were conducted in the Amhara region, two in the Southern Nations, Nationalities, and Peoples' Region (SNNP), and the remaining two in Tigray and Addis Ababa. All studies included in this review employed a cross-sectional design and were published between 2017 and 2023. Details regarding sample size, region, study design, response rate, year of publication, the proportion of health service utilization among fee-waiver beneficiaries, and risk of bias are summarized in the table below (Table 1). Among the studies, the lowest proportion of health service utilization (51.53%) was reported in a study conducted in Tsaeda-Emba District, Tigray Region, Northern Ethiopia [23]. In contrast, the highest proportion (64.88%) was observed in a study conducted in Hawassa, Southern Ethiopia [27]. All included studies were evaluated as having a low risk of bias, with quality scores ranging from 6 to 8. Detailed quality assessment scores for each study are available in the supplemental materials. (Supplemental File 3.)

## Pooled proportion of health service utilization among fee waiver beneficiaries in Ethiopia

The pooled proportion of health service utilization among fee waiver beneficiaries was estimated using a random-effects model, yielding a proportion of 60.57% (95% CI: 58.11–63.04). The analysis revealed a moderate level of heterogeneity across the included studies, with an $I^2$ value of 54.2% and a statistically significant p-value of 0.041, indicating variability in the findings (Fig 2).

**Table 1. Summary of studies included in the systematic review and meta-analysis on health service utilization among fee waiver beneficiaries and its associated factors in Ethiopia, 2024.**

| Author | Publication year | Region | Study design | Sample size | PoHSU (%) | Response rate (%) | Risk of bias | Name of data extractor | Date of data extraction |
|---|---|---|---|---|---|---|---|---|---|
| Amare et al. [23] | 2021 | Tigray | CCS | 163 | 51.53 | 97.60 | Low | BFK | April 15, 2024 |
| Nigusie et al. [21] | 2022 | Amhara | CS | 399 | 62.41 | 98.03 | Low | AHS | April 17, 2024 |
| Chote et al. [26] | 2017 | SNNP | CS | 633 | 59.56 | 100 | Low | BFK | April 20, 2024 |
| Damte et al. [27] | 2022 | SNNP | CS | 581 | 64.88 | 91.35 | Low | BFK | April 22, 2024 |
| Dessie [25] | 2023 | Amhara | CCS | 681 | 58.14 | 97.56 | Low | AHS | April 24, 2024 |
| Jemal et al. [22] | 2022 | Amhara | CS | 545 | 60.92 | 95.80 | Low | AHS | April 27, 2024 |
| Tesfaye [24] | 2017 | Addis Ababa | CS | 438 | 62.33 | 95.20 | Low | BFK | April 25, 2024 |

CS=Cross-sectional, CCS=Comparative Cross-sectional, PoHSU=Proportion Health Service Utilization, SNNP=Southern Nations, Nationalities, and Peoples.

## Subgroup analysis

Subgroup analysis was performed to account for regional variations in the primary studies, as these differences could influence the effect size and contribute to heterogeneity. The analysis revealed that the pooled proportion of health service utilization among fee-waiver beneficiaries was 60.17% (95% CI: 57.72–62.62) in the Amhara region, with low heterogeneity ($I^2 = 5.7\%$, $p = 0.346$). In contrast, studies conducted in the SNNP region reported a higher pooled utilization rate of 62.21% (95% CI: 56.99–67.44), with moderate heterogeneity ($I^2 = 72.8\%$, $p = 0.055$). These findings suggest that studies from the Amhara region demonstrated greater consistency, while those from the SNNP region exhibited heterogeneity (Fig 3).

## Result of publication bias assessment

Publication bias was evaluated using the DOI plot [40] and the Luis Furuya-Kanamori (LFK) index [41]. The results of Egger's test ($p = 0.287$) were not statistically significant ($p > 0.05$), suggesting no substantial evidence of publication bias in the pooled prevalence of depression. Additionally, the LFK index value of −0.98 was within the acceptable range of −1–1, suggesting symmetry in the DOI plot and further supporting the conclusion that publication bias was unlikely in the included studies (Fig 4).

## Sensitivity analysis

A sensitivity analysis was performed to determine if any individual study had an influence on the pooled estimate of health service utilization among fee-waiver beneficiaries. The results showed no significant impact from any single study, confirming the robustness of the overall findings (Fig 5).

## Factors associated with health service utilization among fee waiver beneficiaries in Ethiopia

Factors associated with health service utilization among fee waiver beneficiaries were identified based on the combined results of two or more studies included in this systematic review and meta-analysis. Four studies [21–23,25] contributed data to this analysis. According to the pooled findings from two studies [23,25], fee waiver beneficiaries with a family size of fewer than five members had a 31% lower likelihood of utilizing health services compared to those with a family size of five or more (OR = 0.69; 95% CI: 0.51–0.95). Similarly, the combined results from two other studies [21,25] revealed that beneficiaries with chronic diseases were 4.85 times more likely to utilize health services than those without chronic conditions (OR = 4.85, 95% CI: 1.34–17.56). However, the pooled result of two studies [21,22] showed no statistically

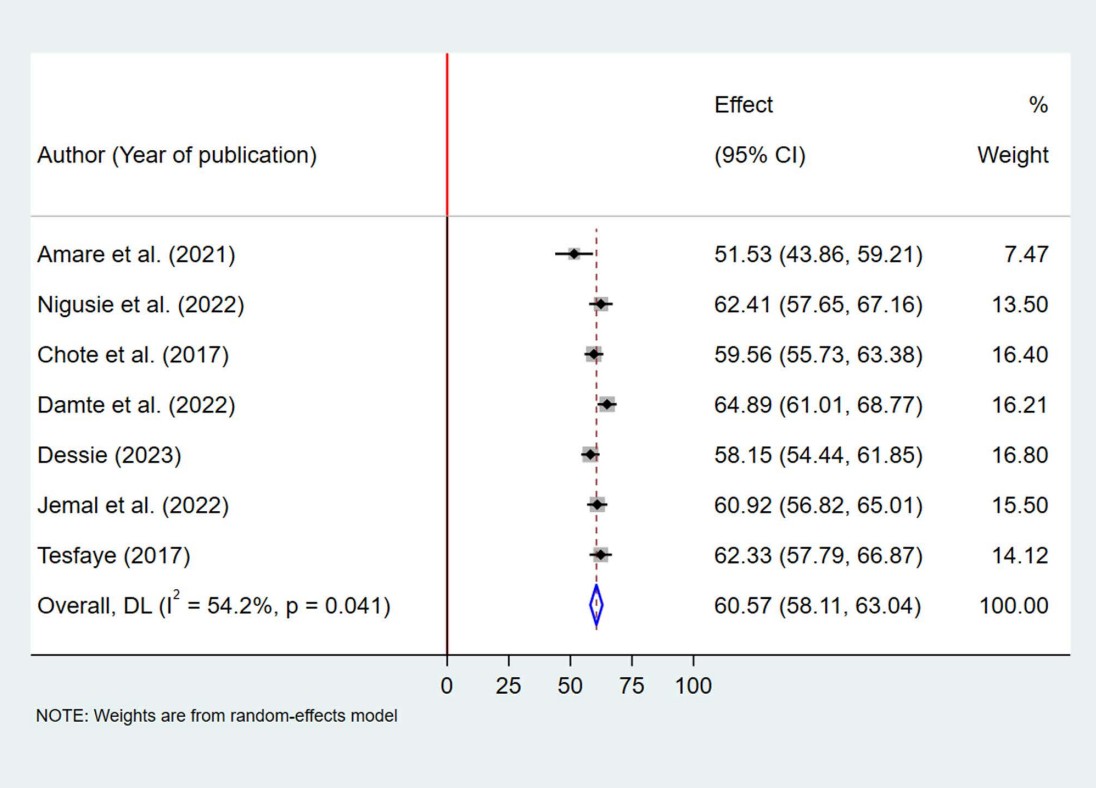

**Fig 2. Forest plot showing the pooled prevalence of health service utilization among fee waiver beneficiaries in Ethiopia, 2024.**

significant association between residence and health service utilization among fee waiver beneficiaries (OR = 1.58; 95% CI: 0.03–71.49), indicating a wide confidence interval and considerable uncertainty in the estimated association (Table 2).

## Discussion

The fee waiver program in Ethiopia is designed to reduce health disparities and ensure equitable access to essential health services by providing financial protection for the most vulnerable populations [28]. This systematic review and meta-analysis examined the prevalence of health service utilization among fee waiver beneficiaries in Ethiopia and identified the factors associated with it. Using a random-effects model, the pooled prevalence of health service utilization among fee waiver beneficiaries in Ethiopia was found to be 60.57%. Family size and presence of chronic diseases were found to be significantly associated with health service in this group.

This finding offers valuable insights into the effectiveness and reach of Ethiopia's fee waiver programs, showcasing progress in improving access to essential healthcare services. The utilization rate indicates that these programs are relatively effective in reducing financial barriers, enabling a substantial portion of the eligible population to seek medical care without the burden of out-of-pocket expenses. This success underscores the programs' critical role in addressing key obstacles to healthcare access in low-income settings [42]. The 60.57% utilization rate also carries significant implications for achieving multiple Sustainable Development Goals (SDGs). It reflects progress toward SDG 3 (Good Health and Well-Being) by enhancing access to healthcare, which improves health outcomes and reduces mortality rates. Furthermore, by alleviating the financial burden of healthcare costs, the fee waiver programs contribute to SDG 1 (No Poverty), preventing low-income households from falling further into poverty. Additionally, the findings support SDG 10 (Reduced

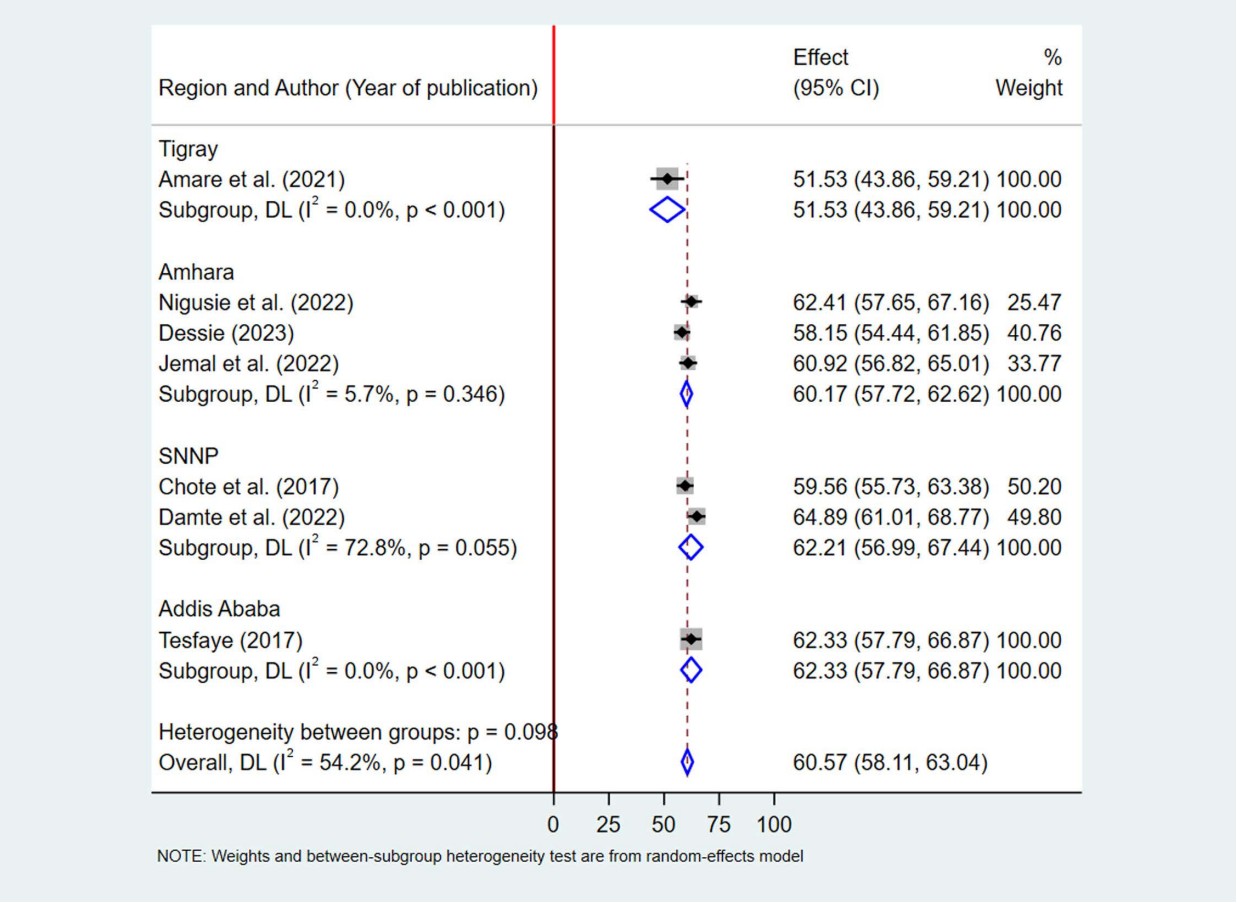

**Fig 3. Subgroup analysis by region showing the pooled prevalence of health service utilization among fee-waiver beneficiaries in Ethiopia, 2024.**

Inequalities) by fostering equitable access to healthcare services and narrowing health disparities across socio-economic groups [43].

The study also revealed that nearly 40% of fee waiver beneficiaries are not utilizing health services, highlighting the need for continued efforts to address existing gaps and reach underserved populations. This underscores the importance of identifying and overcoming the barriers that hinder full utilization, ensuring the program's benefits are accessible to all eligible individuals. By addressing these challenges, the fee waiver programs can further enhance their impact and achieve greater equity in healthcare access. The current finding aligns with a 2017 study conducted in Burkina Faso, which reported that 59.92% of ultra-poor individuals utilized health services [19]. This similarity may be explained by the fact that both Ethiopia and Burkina Faso are classified as low-income countries and face common socio-economic challenges such as limited affordability, predominantly rural populations, inadequate health workforce, and insufficient health sector funding [44]. Moreover, this consistency could be attributed to the fact that, similar to Ethiopia, Burkina Faso has been undertaking healthcare financing reforms since 2002 as part of its efforts to achieve Universal Health Coverage (UHC), with the removal of user fees for the poor being a key component of the reform strategy [45].

Compared to a study conducted in South Korea, the finding of this study is lower, as 93.8% of Medicaid beneficiaries in South Korea utilized health services [10]. This difference might be due to variations in the health systems of the two

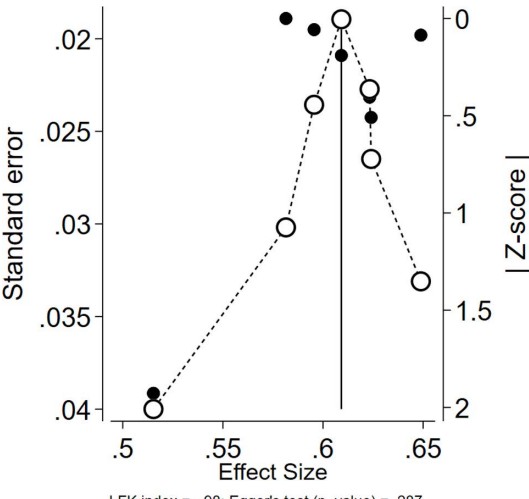

**Fig 4. Assessment of publication bias of included studies on health service utilization among fee-waiver beneficiaries in Ethiopia, 2024.**

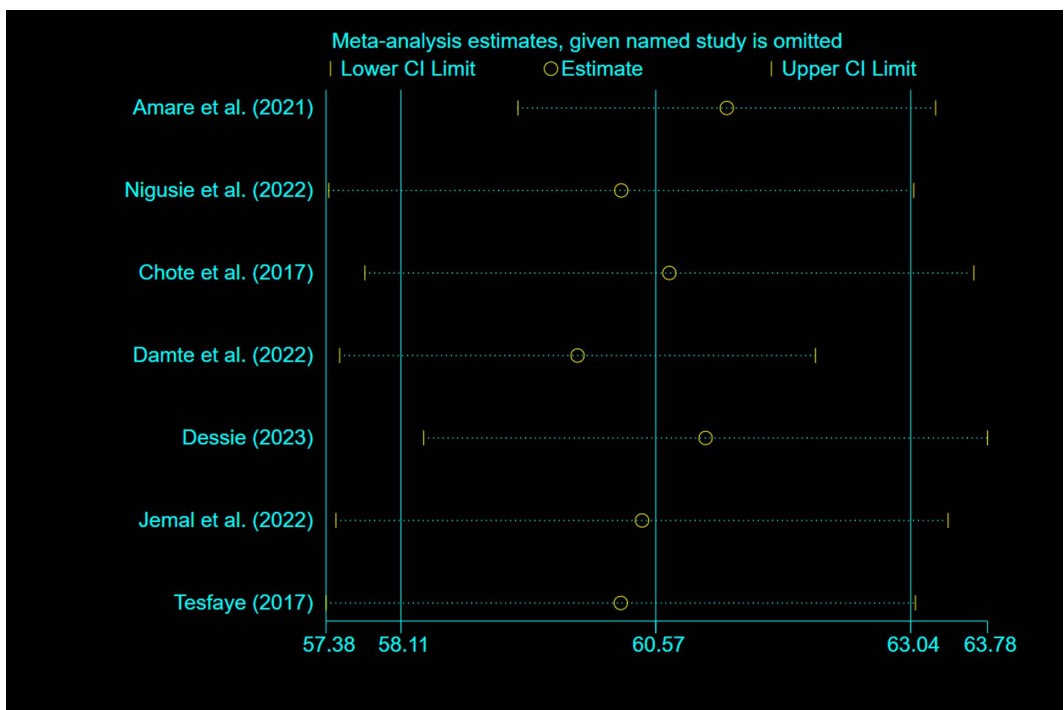

**Fig 5. Sensitivity analysis to determine a single study effect on the pooled prevalence of health service utilization among fee waiver beneficiaries in Ethiopia, 2024.**

countries. South Korea is among the leading countries in healthcare delivery, where most of the population lives close to health facilities and has better access to both essential and specialized care [46].]. In contrast, the Ethiopian healthcare system still faces numerous barriers to accessing basic health services. Furthermore, ongoing war has led to the destruction of healthcare infrastructure and the disruption to essential service delivery [47]. The result is also lower than

**Table 2. A table showing factors associated with health service utilization among fee waiver beneficiaries in Ethiopia, 2024.**

| Variables | Odds ratio with 95% CI | Number of studies | I² with p-value |
|---|---|---|---|
| **Family size** | | | |
| <5 | 0.69 (0.51, 0.95) * | 2 | I²=0.0%, p=0.47 |
| ≥ 5 | 1 | | |
| **Residence** | | | |
| Rural | 1 | 2 | I²=98.9%, p<0.001 |
| Urban | 1.58(0.03, 71.49) | | |
| **Having chronic disease** | | | |
| Yes | 4.85(1.34, 17.56) * | 2 | I²=93.5%, p<0.001 |
| No | | | |

Note: * indicates a significant association; I² shows heterogeneity; the accompanying p-value from Cochran's Q test indicates whether the heterogeneity is statistically significant (p<0.05=significant).

in Madagascar, where 65% of the population utilize health services [48]. Although Madagascar experiences healthcare access issues, it has received more international aid to support its fragile health system, which may have had a favorable impact on access. Another possible explanation is that Ethiopia's relatively larger population may negatively affect the coverage and accessibility of health services [49].

However, the result is higher than that of Uganda, where free healthcare utilization among the poor was 33.67% in 2003 [17]. his may be explained by the fact that, although both countries continue to face healthcare access challenges, a recent study conducted in 2023 in Uganda revealed issues such as a poor referral system, unequal distribution of health facilities, and low government spending [50]. On the other hand, following nationwide healthcare financing reforms in Ethiopia, there have been improvements in revenue generation, risk pooling, and healthcare infrastructure [51].

The findings revealed significant variability in health service utilization rates across the individual studies included in this analysis. The lowest utilization rate was observed in Saesie Tsaeda-Emba District, Tigray Region, at 51.53%, while the highest was recorded in Hawassa, Southern Ethiopia, with a rate of 64.89%. This variability can be attributed to multiple factors affecting healthcare access and utilization. The disparities in utilization rates likely reflect differences in healthcare infrastructure. Urban areas such as Hawassa tend to have better-equipped facilities, a higher number of healthcare providers, and more accessible medical services compared to rural areas like Saesie Tsaeda-Emba District [52]. This underscores the importance of improving healthcare infrastructure in underserved areas to enhance utilization rates.

A subgroup analysis was performed to capture regional variations in the primary studies. The estimated health service utilization rates among fee waiver beneficiaries were 60.17% in the Amhara region and 62.21% in the SNNP region. This analysis provides valuable insights and implications. Despite the geographical and socio-economic differences, both regions exhibited relatively similar utilization rates, suggesting that the fee waiver programs are consistently effective across diverse areas. This indicates that financial barriers to healthcare access are being mitigated to a comparable extent, irrespective of the regional context. However, it remains important to acknowledge and address the unique challenges encountered by each region.

According to the current review, there is a significant association between family size and health service utilization among fee waiver beneficiaries. Families with fewer than five members were less likely to use health services compared to those with five or more members. This can be attributed to several possible explanations, larger families may face a higher cumulative health burden, prompting them to prioritize and seek out healthcare services more frequently. In contrast, smaller families might not feel the same urgency or may perceive a lower overall health risk. As economic factors play a crucial role, larger families might experience greater financial strain, making the relief provided by fee waiver

programs more impactful [53]. This financial assistance could lead to higher utilization rates, as the cost barrier to accessing healthcare is significantly reduced for larger families. Additionally, social support networks also contribute to this dynamic. Larger families often have stronger internal support systems, providing encouragement and practical assistance in accessing healthcare [54]. The collective decision-making process in larger families can lead to prioritizing health needs and ensuring that all members receive necessary medical care [55].

The study also revealed a significant association between chronic diseases and health service utilization. Individuals with chronic conditions were more likely to utilize health services than those without such conditions. Several factors could explain this finding, chronic diseases often require ongoing medical management, including regular check-ups, and various treatments. This necessity for continuous care naturally leads to higher health service utilization among those with chronic conditions. Chronic illnesses are often financially burdensome due to their frequent and long-term healthcare needs [56]. For individuals from lower socioeconomic backgrounds, financial barriers can significantly hinder access to healthcare services, particularly in settings where out-of-pocket payments are the primary means of financing healthcare. Fee waivers alleviate the financial burden associated with these recurring needs, enabling beneficiaries to seek care consistently. Moreover, individuals with chronic conditions often perceive their health problems as more severe and urgent, prompting them to prioritize healthcare services. Preventive and follow-up care, essential for managing chronic conditions and preventing complications, is another key driver of utilization among this group. Fee waivers also support treatment adherence by reducing costs for medications and routine check-ups, making it easier for individuals with chronic illnesses to stay on track with their care plans. Additionally, individuals with chronic diseases are typically more aware of their health status and more proactive in seeking medical attention to manage their condition effectively [57].

The important implication for this finding is that fee waivers not only improve access but also contribute to equitable utilization of healthcare services by reducing disparities between those with and without chronic conditions. Without financial assistance, individuals with chronic illnesses may delay or forego care due to cost concerns, leading to worsening health outcomes and higher costs in the long term. By facilitating access to care, fee waiver programs ensure that individuals with chronic conditions can adhere to treatment plans and benefit from preventive and curative services, ultimately improving their overall health and reducing the burden on the healthcare system.

On the other hand, the current review found no statistically significant association between place of residence and health service utilization among fee waiver beneficiaries. As indicated by the wide confidence interval, this finding suggests considerable uncertainty in the estimated effect size. This imprecision may be attributed to substantial heterogeneity among the included studies. Therefore, further research is needed to determine whether place of residence is truly associated with health service utilization in this population group.

## Strengths and limitations of the study

This study represents the first systematic review and meta-analysis to estimate the pooled prevalence of health service utilization and its associated factors among fee waiver beneficiaries in Ethiopia. An extensive and thorough search was conducted across multiple databases to ensure the inclusion of all relevant studies. However, the study has certain limitations. One limitation is the potential lack of geographical representativeness, as studies from some regions of Ethiopia were not available. Additionally, the moderate heterogeneity observed across studies, reflected in high I² values, may impact the generalizability of the pooled estimates.

## Conclusion

In conclusion, this systematic review and meta-analysis provide valuable insights into health service utilization among fee waiver beneficiaries in Ethiopia. The findings show that a considerable portion of beneficiaries utilizes health services. The analysis also revealed that smaller family sizes (fewer than five members) were significantly associated with reduced health service utilization, while the presence of chronic diseases was significantly associated with increased utilization. Importantly, all included studies demonstrated a low risk of bias, with no significant evidence of publication bias detected.

These findings underscore the need for targeted interventions to address factors influencing health service utilization among vulnerable populations, particularly those with smaller family sizes and chronic health conditions.

## Supporting information

**Supplemental File 1. All databases search strategy detail for systematic review and meta-analysis to estimate the prevalence of health service utilization and associated factors among fee waiver beneficiaries in Ethiopia, 2024.**
(DOCX)

**Supplemental File 2. List of excluded studies from the systematic review and meta-analysis of health service utilization and associated factors among fee waiver beneficiaries in Ethiopia, 2024.**
(DOCX)

**Supplemental File 3. Quality appraisal results of included analytical cross-sectional studies in Ethiopia, Using Joanna Briggs Institute (JBI) quality appraisal checklist for systematic review and meta-analysis.**
(DOCX)

**Supplemental File 4. PRISMA_2020_checklist for systematic review and meta-analysis to estimate the prevalence of health service utilization and associated factors among fee waiver beneficiaries in Ethiopia, 2024.**
(DOCX)

**Supplemental File 5. PLOS ONE Clinical Studies Checklist for systematic review and meta-analysis to estimate the prevalence of health service utilization and associated factors among fee waiver beneficiaries in Ethiopia, 2024.**
(DOCX)

## Acknowledgments

We extend our gratitude to Samara University for granting free access to the internet and library resources, as well as to all the authors of the primary studies referenced in this work.

## Author contributions

**Conceptualization:** Bizunesh Fantahun Kase.

**Data curation:** Bizunesh Fantahun Kase, Hiwot Altaye Asebe.

**Formal analysis:** Bizunesh Fantahun Kase, Etsay Woldu Anbesu, Dejen Kahsay Asgedom, Nuru Mohammed Hussen, Abdu Hailu Shibeshi.

**Investigation:** Bizunesh Fantahun Kase, Abdulkerim Hassen Moloro.

**Methodology:** Bizunesh Fantahun Kase, Abubeker Alebachew Seid.

**Software:** Bizunesh Fantahun Kase, Aragaw Asfaw Hasen, Abdu Hailu Shibeshi.

**Writing – original draft:** Bizunesh Fantahun Kase.

**Writing – review & editing:** Bizunesh Fantahun Kase, Molla Getie Mehari, Kassaye Getaneh Arge.

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
