## [Decision Letter · Decision Letter 0]

Dear Dr. Kase,

Thank you for submitting your manuscript to PLOS ONE. After careful consideration, we feel that it has merit but does not fully meet PLOS ONE’s publication criteria as it currently stands. Therefore, we invite you to submit a revised version of the manuscript that addresses the points raised during the review process.

**ACADEMIC EDITOR: **

The finding on residence is missing from the abstract results. The introduction can be improved by consolidating the first two paragraphs into one and highlighting fee waiver as a pathway to universal health coverage and equitable access to health care. Lines 50-51 focus on rural and remote areas. The authors should also highlight urban inequities in access to healthcare. In line 67-71, the authors describe the fee waiver policy in Ethiopia, but this case description needs expansion. In line 72-75, the authors analyse the knowledge gap, but this gap analysis is not comprehensive. The authors should review the pertinent literature on how fee waiver policies influence health service utilization elsewhere to enrich the gap analysis. The authors should also be more systematic in the Discussion section. Discuss the pooled prevalence, regional and sub-group variations first, and then the factors affacting health service utilisation. Notably, the findings about place of residence was not discussed. So, the authors should include a discussion of the urban-rural disparity in health service utilisation.   

We look forward to receiving your revised manuscript.

Kind regards,

Daniel Chukwuemeka Ogbuabor, Ph.D., M.D.

Academic Editor

PLOS ONE

Journal Requirements:

2. As required by our policy on Data Availability, please ensure your manuscript or supplementary information includes the following:

3. We note you have included a table to which you do not refer in the text of your manuscript. Please ensure that you refer to Table 2 in your text; if accepted, production will need this reference to link the reader to the Table.

Additional Editor Comments :

The finding on residence is missing from the abstract results. The introduction can be improved by consolidating the first two paragraphs into one and highlighting fee waiver as a pathway to universal health coverage and equitable access to health care. Lines 50-51 focus on rural and remote areas. The authors should also highlight urban inequities in access to healthcare. In line 67-71, the authors describe the fee waiver policy in Ethiopia, but this case description needs expansion. In line 72-75, the authors analyse the knowledge gap, but this gap analysis is not comprehensive. The authors should review the pertinent literature on how fee waiver policies influence health service utilization elsewhere to enrich the gap analysis. The authors should also be more systematic in the Discussion section. Discuss the pooled prevalence, regional and sub-group variations first, and then the factors affacting health service utilisation. Notably, the findings about place of residence was not discussed. So, the authors should include a discussion of the urban-rural disparity in health service utilisation.

Reviewers' comments:

Reviewer's Responses to Questions

**Comments to the Author**

1. Is the manuscript technically sound, and do the data support the conclusions?

Reviewer #1: Yes

Reviewer #2: Yes

2. Has the statistical analysis been performed appropriately and rigorously?

Reviewer #1: Yes

Reviewer #2: Yes

3. Have the authors made all data underlying the findings in their manuscript fully available?

Reviewer #1: Yes

Reviewer #2: Yes

4. Is the manuscript presented in an intelligible fashion and written in standard English?

Reviewer #1: Yes

Reviewer #2: Yes

Reviewer #1: The manuscript is well-structured, and the data presented robustly supports the conclusions drawn. All data is fully accessible, and the statistical analysis has been conducted with appropriate rigor. While the manuscript is generally clear and written in standard English, several grammatical errors are present throughout. These should be carefully reviewed and revised to ensure the precise and accurate communication of the research findings, thereby enhancing clarity and reinforcing the author's expertise.

Reviewer #2: Abstract: no need to put again the percentage at the conclusion. Re-paraphrase the conclusion as the findings. Line 41-42 can leave there as for the recommendation of the study.

Introduction: perfect and clear

Methods: okey

Results: Line 208 (OR = 0.69; 95% CI: 0.51, 0.95), please check for the notation at the CI. I think it supposes to be minus (-) to replace coma (,).

Discussion: Okey

Conclusion: Line 285-286 can be deleted for this sentence “Based on seven primary studies with a total of 11,488 participants, the pooled prevalence of health service utilization was 60.57%, indicating moderate utilization levels” as it has already mentioned at the results and discussion.

**Do you want your identity to be public for this peer review?** For information about this choice, including consent withdrawal, please see our Privacy Policy

Reviewer #1: **Yes: ** Dr. Nicole Bell-Rogers FNP-C, PMHNP-BC, RN

Reviewer #2: **Yes: ** Ni Ketut Aryastami

---

## [Author Response · Author response to Decision Letter 1]

12 May 2025

Point by Point Response to the Editor, and Reviewers Comments

Title: Health service utilization and associated factors among fee waiver beneficiaries in Ethiopia: Systematic review and meta-analysis

Manuscript ID: PONE-D-25-04792

Date: May, 2025

Subject: Revision of the manuscript

Dear Editor and Reviewers, we sincerely appreciate the time and effort you have dedicated to reviewing our manuscript and providing insightful comments. We found your feedback to be constructive and vital in improving the quality of our work. In response, we have carefully revised the manuscript and addressed all the questions and comments raised by the editors and reviewers. As requested, we have included a point-by-point response to your feedback below.

Thank you once again for your valuable input.

Sincerely,

On behalf of Co-authors.

Bizunesh Fantahun Kase (Corresponding Author)

Journal Requirements

Response: Thank you for the reminder. We have reviewed and updated the manuscript and all associated files to ensure they meet PLOS ONE's style requirements.

2. As required by our policy on Data Availability, please ensure your manuscript or supplementary information includes the following:

Response: Thank you for your detailed and helpful instructions regarding data availability. We have ensured that all required information is included in the revised manuscript and supplementary materials.

3. We note you have included a table to which you do not refer in the text of your manuscript. Please ensure that you refer to Table 2 in your text; if accepted, production will need this reference to link the reader to the Table.

Response: Thank you for your helpful comment. We have now referenced Table 2 in the text of the manuscript.

Response: Thank you for your helpful suggestion. We have included captions for all Supporting Information files at the end of the revised manuscript.

Response: Thank you for your careful review. We have thoroughly reviewed the reference list and can confirm that no retracted references are included in the manuscript. All references cited are current and appropriate for the study.

Editor's Comments and Authors’ Response

1. The finding on residence is missing from the abstract results.

Response: We appreciate your careful review of our manuscript. In response to your comment, we have now included the finding on residence in the abstract results section.

2. The introduction can be improved by consolidating the first two paragraphs into one and highlighting fee waiver as a pathway to universal health coverage and equitable access to health care.

Response: Thank you for your valuable comment. We have addressed this point in the revised manuscript as requested.

3. Lines 50-51 focus on rural and remote areas. The authors should also highlight urban inequities in access to healthcare.

Response: We appreciate the suggestion to address urban inequities in access to healthcare. In response, we have revised the manuscript to highlight urban inequalities alongside those in rural areas.

4. In line 67-71, the authors describe the fee waiver policy in Ethiopia, but this case description needs expansion.

Response: Thank you for your valuable comment. We have now expanded the description to provide greater clarity and context.

5. In line 72-75, the authors analyse the knowledge gap, but this gap analysis is not comprehensive. The authors should review the pertinent literature on how fee waiver policies influence health service utilization elsewhere to enrich the gap analysis.

Response: Thank you for your insightful suggestion. We have addressed this comment by expanding the gap analysis in the revised manuscript. We have reviewed and incorporated relevant literature on the influence of fee waiver policies on health service utilization in other settings.

6. The authors should also be more systematic in the Discussion section. Discuss the pooled prevalence, regional and sub-group variations first, and then the factors affecting health service utilisation.

Response: Thank you for your constructive feedback. We have considered your suggestion and revised the Discussion section accordingly.

7. Notably, the findings about place of residence was not discussed. So, the authors should include a discussion of the urba--n-rural disparity in health service utilisation.

Response: Thank you for highlighting this important point. We have addressed your comment by including a discussion of the urban-rural disparity in health service utilization in the revised manuscript.

Reviewer #1

1. The manuscript is well-structured, and the data presented robustly supports the conclusions drawn. All data is fully accessible, and the statistical analysis has been conducted with appropriate rigor. While the manuscript is generally clear and written in standard English, several grammatical errors are present throughout. These should be carefully reviewed and revised to ensure the precise and accurate communication of the research findings, thereby enhancing clarity and reinforcing the author's expertise.

Response: Thank you for your positive feedback. We have carefully reviewed the manuscript and revised the grammatical errors to ensure accurate communication of our findings.

Reviewer #2

1. Abstract: no need to put again the percentage at the conclusion. Re-paraphrase the conclusion as the findings.

Response: We appreciate your suggestion. We have revised the conclusion section of the abstract by removing the repeated percentage and rephrasing it.

2. Line 41-42 can leave there as for the recommendation of the study.

Response: Thank you for your valuable comment. We included the statement in the abstract in accordance with the PLOS ONE submission guidelines, which recommend summarizing the most important results and their significance. Additionally, previously published research articles in the journal also report recommendations under the conclusion section of the abstract. However, if our explanation does not fully address your concern, we are willing to revise the manuscript based on your recommendation.

3. Introduction: perfect and clear

Response: Thank you for the positive feedback on the introduction.

4. Methods: okey

Response: We appreciate your positive assessment.

5. Results: Line 208 (OR = 0.69; 95% CI: 0.51, 0.95), please check for the notation at the CI. I think it supposes to be minus (-) to replace coma (,).

Response: Thank you for your comment. We have reviewed and corrected the notation in the confidence interval.

6. Discussion: Okey

Response: Thank you for the positive feedback

7. Conclusion: Line 285-286 can be deleted for this sentence “Based on seven primary studies with a total of 11,488 participants, the pooled prevalence of health service utilization was 60.57%, indicating moderate utilization levels” as it has already mentioned at the results and discussion.

Response: Thank you for your insightful comment. We agree with your suggestion and have deleted the sentence to avoid redundancy.

---

## [Editor Report · Decision Letter 1]

Dear Dr. Kase,

In the Background, line 57-61 of the revised manuscript focused on the pertinent literature. However, four papers are few, and do not represent the existing scholarship in this area. Equally, the ommitted the review of the factors associated with utilization of free care services.In the Discussion section, the authors should expunge the last sentence of the first paragraph of the Discussion section. In its place, they should summarize the key findings that were subsequently discussed.In the Discussion, the authors completely ommitted comparing the study's findings with existing literature. It is the practice to compare study findings to existing literature and explain any variations. Doing this will enrich the discussion.

We look forward to receiving your revised manuscript.

Kind regards,

Daniel Chukwuemeka Ogbuabor, Ph.D., M.D.

Academic Editor

PLOS ONE

Journal Requirements:

Additional Editor Comments:

I commend the authors for addressing most the comments. Nevertheless, there are a few important concerns that must be addressed to bring the manuscript to a publishable level. In the Background, line 57-61 of the revised manuscript focused on the pertinent literature. However, four papers are few, and do not represent the existing scholarship in this area. Equally, the ommitted the review of the factors associated with utilization of free care services.

In the Discussion section, the authors should expunge the last sentence of the first paragraph of the Discussion section. In its place, they should summarize the key findings that were subsequently discussed.

The authors completely ommitted comparing the study's findings with existing literature. It is the practice to compare study findings to existing literature and explain any variations. Doing this will enrich the discussion.

---

## [Author Response · Author response to Decision Letter 2]

17 May 2025

Point by Point Response to the Editor, and Reviewers Comments

Title: Health service utilization and associated factors among fee waiver beneficiaries in Ethiopia: Systematic review and meta-analysis

Manuscript ID: PONE-D-25-04792

Date: May, 2025

Subject: Revision of the manuscript

Dear Editor and Reviewers, we sincerely appreciate the time and effort you have dedicated to reviewing our manuscript and providing insightful comments. We found your feedback to be constructive and vital in improving the quality of our work. In response, we have carefully revised the manuscript and addressed all the questions and comments raised by the editors and reviewers. As requested, we have included a point-by-point response to your feedback below.

Thank you once again for your valuable input.

Sincerely,

On behalf of Co-authors.

Bizunesh Fantahun Kase (Corresponding Author)

Journal Requirements

Response: Thank you for the reminder. We have reviewed and updated the manuscript and all associated files to ensure they meet PLOS ONE's style requirements.

2. As required by our policy on Data Availability, please ensure your manuscript or supplementary information includes the following:

Response: Thank you for your detailed and helpful instructions regarding data availability. We have ensured that all required information is included in the revised manuscript and supplementary materials.

3. We note you have included a table to which you do not refer in the text of your manuscript. Please ensure that you refer to Table 2 in your text; if accepted, production will need this reference to link the reader to the Table.

Response: Thank you for your helpful comment. We have now referenced Table 2 in the text of the manuscript.

Response: Thank you for your helpful suggestion. We have included captions for all Supporting Information files at the end of the revised manuscript.

Response: Thank you for your careful review. We have thoroughly reviewed the reference list and can confirm that no retracted references are included in the manuscript. All references cited are current and appropriate for the study.

Editor's Comments and Authors’ Response

1. The finding on residence is missing from the abstract results.

Response: We appreciate your careful review of our manuscript. In response to your comment, we have now included the finding on residence in the abstract results section.

2. The introduction can be improved by consolidating the first two paragraphs into one and highlighting fee waiver as a pathway to universal health coverage and equitable access to health care.

Response: Thank you for your valuable comment. We have addressed this point in the revised manuscript as requested.

3. Lines 50-51 focus on rural and remote areas. The authors should also highlight urban inequities in access to healthcare.

Response: We appreciate the suggestion to address urban inequities in access to healthcare. In response, we have revised the manuscript to highlight urban inequalities alongside those in rural areas.

4. In line 67-71, the authors describe the fee waiver policy in Ethiopia, but this case description needs expansion.

Response: Thank you for your valuable comment. We have now expanded the description to provide greater clarity and context.

5. In line 72-75, the authors analyse the knowledge gap, but this gap analysis is not comprehensive. The authors should review the pertinent literature on how fee waiver policies influence health service utilization elsewhere to enrich the gap analysis.

Response: Thank you for your insightful suggestion. We have addressed this comment by expanding the gap analysis in the revised manuscript. We have reviewed and incorporated relevant literature on the influence of fee waiver policies on health service utilization in other settings.

6. The authors should also be more systematic in the Discussion section. Discuss the pooled prevalence, regional and sub-group variations first, and then the factors affecting health service utilisation.

Response: Thank you for your constructive feedback. We have considered your suggestion and revised the Discussion section accordingly.

7. Notably, the findings about place of residence was not discussed. So, the authors should include a discussion of the urba--n-rural disparity in health service utilisation.

Response: Thank you for highlighting this important point. We have addressed your comment by including a discussion of the urban-rural disparity in health service utilization in the revised manuscript.

8. In the Background, line 57-61 of the revised manuscript focused on the pertinent literature. However, four papers are few, and do not represent the existing scholarship in this area. Equally, the ommitted the review of the factors associated with utilization of free care services.

Response: Thank you for the comment. We have expanded the Background section to include additional relevant literature and add information on key factors associated with the utilization of health services among fee waiver beneficiaries as suggested.

9. In the Discussion section, the authors should expunge the last sentence of the first paragraph of the Discussion section. In its place, they should summarize the key findings that were subsequently discussed.

Response: Thank you. We have removed the sentence and replaced it with a summary of the key findings as advised.

10. In the Discussion, the authors completely ommitted comparing the study's findings with existing literature. It is the practice to compare study findings to existing literature and explain any variations. Doing this will enrich the discussion.

Response: Thank you for the insightful comment. We have now expanded the Discussion section to compare our findings with existing literature.

Reviewer #1

1. The manuscript is well-structured, and the data presented robustly supports the conclusions drawn. All data is fully accessible, and the statistical analysis has been conducted with appropriate rigor. While the manuscript is generally clear and written in standard English, several grammatical errors are present throughout. These should be carefully reviewed and revised to ensure the precise and accurate communication of the research findings, thereby enhancing clarity and reinforcing the author's expertise.

Response: Thank you for your positive feedback. We have carefully reviewed the manuscript and revised the grammatical errors to ensure accurate communication of our findings.

Reviewer #2

1. Abstract: no need to put again the percentage at the conclusion. Re-paraphrase the conclusion as the findings.

Response: We appreciate your suggestion. We have revised the conclusion section of the abstract by removing the repeated percentage and rephrasing it.

2. Line 41-42 can leave there as for the recommendation of the study.

Response: Thank you for your valuable comment. We included the statement in the abstract in accordance with the PLOS ONE submission guidelines, which recommend summarizing the most important results and their significance. Additionally, previously published research articles in the journal also report recommendations under the conclusion section of the abstract. However, if our explanation does not fully address your concern, we are willing to revise the manuscript based on your recommendation.

3. Introduction: perfect and clear

Response: Thank you for the positive feedback on the introduction.

4. Methods: okey

Response: We appreciate your positive assessment.

5. Results: Line 208 (OR = 0.69; 95% CI: 0.51, 0.95), please check for the notation at the CI. I think it supposes to be minus (-) to replace coma (,).

Response: Thank you for your comment. We have reviewed and corrected the notation in the confidence interval.

6. Discussion: Okey

Response: Thank you for the positive feedback

7. Conclusion: Line 285-286 can be deleted for this sentence “Based on seven primary studies with a total of 11,488 participants, the pooled prevalence of health service utilization was 60.57%, indicating moderate utilization levels” as it has already mentioned at the results and discussion.

Response: Thank you for your insightful comment. We agree with your suggestion and have deleted the sentence to avoid redundancy.

---

## [Editor Report · Decision Letter 2]

Dear Dr. Bizunesh Fantahun Kase,

Thank you for submitting your manuscript to PLOS ONE. After careful consideration, we feel that it has merit but does not fully meet PLOS ONE’s publication criteria as it currently stands. Therefore, we invite you to submit a revised version of the manuscript that addresses the points raised during the review process.

**ACADEMIC EDITOR: Please insert comments here and delete this placeholder text when finished.**

The study set significance level at p-value less than 0.05. On page 20, Table 2 indicates that family size with p-value = 0.47 is not significant, while urban residence and having chronic disease significantly influenced health service utilisation (p-value <0.001). However, the abstract, results, and discussion sections indicate that residence do not significantly influence health service utilization, while family size significantly influenced health utilization. The discrepancies between Table 2 and the narratives need clarification from the authors, as these could affect the abstract, results and discussion.

We look forward to receiving your revised manuscript.

Kind regards,

Daniel Chukwuemeka Ogbuabor, Ph.D., M.D.

Academic Editor

PLOS ONE

Journal Requirements:

Additional Editor Comments (if provided):

The study set significance level at p-value less than 0.05. On page 20, Table 2 indicates that family size with p-value = 0.47 is not significant, while urban residence and having chronic disease significantly influenced health service utilisation (p-value <0.001). However, the abstract, results, and discussion sections indicate that residence do not significantly influence health service utilization, while family size significantly influenced health utilization. The discrepancies between Table 2 and the narratives need clarification from the authors, as these could affect the abstract, results and discussion.

---

## [Author Response · Author response to Decision Letter 3]

22 May 2025

Point by Point Response to the Editor, and Reviewers Comments

Title: Health service utilization and associated factors among fee waiver beneficiaries in Ethiopia: Systematic review and meta-analysis

Manuscript ID: PONE-D-25-04792

Date: May, 2025

Subject: Revision of the manuscript

Dear Editor and Reviewers, we sincerely appreciate the time and effort you have dedicated to reviewing our manuscript and providing insightful comments. We found your feedback to be constructive and vital in improving the quality of our work. In response, we have carefully revised the manuscript and addressed all the questions and comments raised by the editors and reviewers. As requested, we have included a point-by-point response to your feedback below.

Thank you once again for your valuable input.

Sincerely,

On behalf of Co-authors.

Bizunesh Fantahun Kase (Corresponding Author)

Journal Requirements

Response: Thank you for the reminder. We have reviewed and updated the manuscript and all associated files to ensure they meet PLOS ONE's style requirements.

2. As required by our policy on Data Availability, please ensure your manuscript or supplementary information includes the following:

Response: Thank you for your detailed and helpful instructions regarding data availability. We have ensured that all required information is included in the revised manuscript and supplementary materials.

3. We note you have included a table to which you do not refer in the text of your manuscript. Please ensure that you refer to Table 2 in your text; if accepted, production will need this reference to link the reader to the Table.

Response: Thank you for your helpful comment. We have now referenced Table 2 in the text of the manuscript.

Response: Thank you for your helpful suggestion. We have included captions for all Supporting Information files at the end of the revised manuscript.

Response: Thank you for your careful review. We have thoroughly reviewed the reference list and can confirm that no retracted references are included in the manuscript. All references cited are current and appropriate for the study.

Editor's Comments and Authors’ Response

1. The finding on residence is missing from the abstract results.

Response: We appreciate your careful review of our manuscript. In response to your comment, we have now included the finding on residence in the abstract results section.

2. The introduction can be improved by consolidating the first two paragraphs into one and highlighting fee waiver as a pathway to universal health coverage and equitable access to health care.

Response: Thank you for your valuable comment. We have addressed this point in the revised manuscript as requested.

3. Lines 50-51 focus on rural and remote areas. The authors should also highlight urban inequities in access to healthcare.

Response: We appreciate the suggestion to address urban inequities in access to healthcare. In response, we have revised the manuscript to highlight urban inequalities alongside those in rural areas.

4. In line 67-71, the authors describe the fee waiver policy in Ethiopia, but this case description needs expansion.

Response: Thank you for your valuable comment. We have now expanded the description to provide greater clarity and context.

5. In line 72-75, the authors analyse the knowledge gap, but this gap analysis is not comprehensive. The authors should review the pertinent literature on how fee waiver policies influence health service utilization elsewhere to enrich the gap analysis.

Response: Thank you for your insightful suggestion. We have addressed this comment by expanding the gap analysis in the revised manuscript. We have reviewed and incorporated relevant literature on the influence of fee waiver policies on health service utilization in other settings.

6. The authors should also be more systematic in the Discussion section. Discuss the pooled prevalence, regional and sub-group variations first, and then the factors affecting health service utilisation.

Response: Thank you for your constructive feedback. We have considered your suggestion and revised the Discussion section accordingly.

7. Notably, the findings about place of residence was not discussed. So, the authors should include a discussion of the urba--n-rural disparity in health service utilisation.

Response: Thank you for highlighting this important point. We have addressed your comment by including a discussion of the urban-rural disparity in health service utilization in the revised manuscript.

8. In the Background, line 57-61 of the revised manuscript focused on the pertinent literature. However, four papers are few, and do not represent the existing scholarship in this area. Equally, the ommitted the review of the factors associated with utilization of free care services.

Response: Thank you for the comment. We have expanded the Background section to include additional relevant literature and add information on key factors associated with the utilization of health services among fee waiver beneficiaries as suggested.

9. In the Discussion section, the authors should expunge the last sentence of the first paragraph of the Discussion section. In its place, they should summarize the key findings that were subsequently discussed.

Response: Thank you. We have removed the sentence and replaced it with a summary of the key findings as advised.

10. In the Discussion, the authors completely ommitted comparing the study's findings with existing literature. It is the practice to compare study findings to existing literature and explain any variations. Doing this will enrich the discussion.

Response: Thank you for the insightful comment. We have now expanded the Discussion section to compare our findings with existing literature.

11. The study set significance level at p-value less than 0.05. On page 20, Table 2 indicates that family size with p-value = 0.47 is not significant, while urban residence and having chronic disease significantly influenced health service utilisation (p-value <0.001). However, the abstract, results, and discussion sections indicate that residence do not significantly influence health service utilization, while family size significantly influenced health utilization. The discrepancies between Table 2 and the narratives need clarification from the authors, as these could affect the abstract, results and discussion.

Response: Thank you for your valuable observation, and we sincerely apologize for the confusion caused. Regarding the p value reported in table two, we would like to clarify that I² indicates the percentage of heterogeneity, and the accompanying p-value is from Cochran’s Q test, which assesses the statistical significance of this heterogeneity. A p-value < 0.05 suggests significant heterogeneity. This explanation has now been clearly stated in the footnote of Table 2 in the revised manuscript.

Reviewer #1

1. The manuscript is well-structured, and the data presented robustly supports the conclusions drawn. All data is fully accessible, and the statistical analysis has been conducted with appropriate rigor. While the manuscript is generally clear and written in standard English, several grammatical errors are present throughout. These should be carefully reviewed and revised to ensure the precise and accurate communication of the research findings, thereby enhancing clarity and reinforcing the author's expertise.

Response: Thank you for your positive feedback. We have carefully reviewed the manuscript and revised the grammatical errors to ensure accurate communication of our findings.

Reviewer #2

1. Abstract: no need to put again the percentage at the conclusion. Re-paraphrase the conclusion as the findings.

Response: We appreciate your suggestion. We have revised the conclusion section of the abstract by removing the repeated percentage and rephrasing it.

2. Line 41-42 can leave there as for the recommendation of the study.

Response: Thank you for your valuable comment. We included the statement in the abstract in accordance with the PLOS ONE submission guidelines, which recommend summarizing the most important results and their significance. Additionally, previously published research articles in the journal also report recommendations under the conclusion section of the abstract. However, if our explanation does not fully address your concern, we are willing to revise the manuscript based on your recommendation.

3. Introduction: perfect and clear

Response: Thank you for the positive feedback on the introduction.

4. Methods: okey

Response: We appreciate your positive assessment.

5. Results: Line 208 (OR = 0.69; 95% CI: 0.51, 0.95), please check for the notation at the CI. I think it supposes to be minus (-) to replace coma (,).

Response: Thank you for your comment. We have reviewed and corrected the notation in the confidence interval.

6. Discussion: Okey

Response: Thank you for the positive feedback

7. Conclusion: Line 285-286 can be deleted for this sentence “Based on seven primary studies with a total of 11,488 participants, the pooled prevalence of health service utilization was 60.57%, indicating moderate utilization levels” as it has already mentioned at the results and discussion.

Response: Thank you for your insightful comment. We agree with your suggestion and have deleted the sentence to avoid redundancy.

---

## [Editor Report · Decision Letter 3]

Health service utilization and associated factors among fee waiver beneficiaries in Ethiopia: Systematic review and meta-analysis

PONE-D-25-04792R3

Dear Dr. Bizunesh Fantahun Kase,

We’re pleased to inform you that your manuscript has been judged scientifically suitable for publication and will be formally accepted for publication once it meets all outstanding technical requirements.

Kind regards,

Daniel Chukwuemeka Ogbuabor, Ph.D., M.D.

Academic Editor

PLOS ONE

Additional Editor Comments (optional):

The paper is accepted but the authors should insert the p-values for the odd ratios in Table 2. The p-values for the index of heterogeneity of the papers does not replace those of the odd ratios.
---

## [Editor Report · Acceptance letter]

PONE-D-25-04792R3

PLOS ONE

Dear Dr. Kase,

I'm pleased to inform you that your manuscript has been deemed suitable for publication in PLOS ONE. Congratulations! Your manuscript is now being handed over to our production team.

Kind regards,

on behalf of

Dr. Daniel Chukwuemeka Ogbuabor

Academic Editor

PLOS ONE